# Virulence signature of the endemic vancomycin-resistant *Enterococcus faecium* clones in Denmark, 2015-2023

Ingrid Maria Cecilia Rubin,[1,2,3] Kasper Thystrup Karstensen,[1] Mikkel Lindegaard,[1] Kristin Hegstad,[4,5] Ana R. Freitas,[6,7,8] Anette M. Hammerum,[1] Louise Roer[1]

**ABSTRACT** Since 2012, Denmark has seen a significant rise in vancomycin-resistant *Enterococcus faecium* (VREfm) cases, mirroring trends in other countries, though exceptions occurred during the coronavirus disease 2019 (COVID-19) lockdown and between 2022 and 2023. This rise has been accompanied by ongoing changes in the endemic VREfm clones, reflecting the species' high genetic plasticity. VREfm rapidly acquires plasmids and mobile genetic elements, enriching it with putative virulence markers (PVMs), including surface proteins, pili, and factors encoding biofilm production and adhesion. L. Roer, H. Kaya, A.P. Tedim, C. Novias, et al. (Microbiol Spectr 12:e0372423, 2024, https://doi.org/10.1128/spectrum.03724-23) released a database of 27 PVMs for *Enterococcus faecium* and *Enterococcus lactis*. In this study, we examined 516 VREfm bloodstream isolates from 2015 to 2023, identifying eight new putative virulence genes added to the database, bringing the total to 35 PVMs. Using whole-genome sequencing (WGS) and single linkage clustering, we identified six dominant VREfm clusters: ST80-CT14 *vanA*, ST117-CT24 *vanA*, ST203-CT859 *vanA*, ST1421-CT1134 *vanA*, ST117-CT36 *vanB*, and ST80-CT2406 *vanB*. We observed significant differences in the distribution of PVMs, particularly in pilin, gene clusters (PGC-1, PGC-2, and PGC-4), genes involved in carbohydrate metabolism (e.g., *orf1481*, *ccpA*), and biofilm production (e.g., *hylEfm*, *capD*, *lysM3*). These differences could explain variability in pathogenicity, metabolism, and adaptation to stress, contributing to shifts in endemic clones.

**IMPORTANCE** The newly developed and now updated database, VirulenceFinder, features 35 potential virulence markers for E. *faecium* and E. *lactis*, is highly scalable and provides a valuable tool for the in-depth analysis of closely related species using whole-genome sequencing (WGS) data. It holds considerable promise for a range of public health applications, such as hospital outbreak investigations, surveillance, and assessment of pathogenicity of bacterial species.

**KEYWORDS** virulence markers, virulenceFinder, VRE, *Enterococcus faecium*

*E*nterococcus faecium (*Efm*) has emerged as one of the major causes of hospital-associated infections around the globe, with a range of clinical infections, including urinary tract infections, bacteremia, infections of indwelling medical devices, and gastrointestinal infections (1–4). The emergence of vancomycin-resistant *E. faecium* (VREfm) in the late 1980s was of special concern, as vancomycin had been a first-line treatment of *E. faecium* infections and considered a last-resort antibiotic (2, 3). Since VREfm is now included in the WHO's list of priority pathogens and accounts for a significant share of hospital-acquired infections globally, this emphasizes the urgent need for effective strategies and surveillance to manage these infections (5–8).

In *Efm*, vancomycin resistance is conferred by 10 different *van* clusters, each composed of two to three operons, with the *vanA* and *vanB* genes being the most

Address correspondence to Louise Roer, loro@ssi.dk.

The authors declare no conflict of interest.

The data presented in this article were presented as a poster (01298) at ESCMID Global, April 2025, in Vienna, Austria.

common in human clinical VREfm isolates. Historically, and with the onset of whole genome sequencing (WGS), *Efm* has been separated into two major groups, clade A and clade B; however, late studies show that isolates in clade B share more than 97% nucleotide identity with *Enterococcus lactis,* and therefore it was proposed that this clade should be re-named *Enterococcus lactis (Elts)* (9–13). Furthermore, with the rising number of sequenced isolates, more detail into the phylogeny of *Efm* has demonstrated that clade A is further subdivided into clade A1, mainly composed of hospital isolates and non-A1 clade, mainly composed of animal and human commensal isolates (12–15). Chiefly composed of clinical isolates that prevail in hospital environments, it is unsurprising that strains of clade A1 exhibit larger genomes, elevated mutation rates, and a greater number of antibiotic resistance genes, along with an increased presence of mobile genetic elements (MGEs) compared with strains from either the non-A1 or B clades (12).

Despite not being considered particularly virulent, enterococci host several traits that are considered virulence factors, this including genes encoding surface proteins, pili, and secreted virulence factors relevant to the adhesion to host tissues, biofilm production, and pathogenesis (16). The most studied putative virulence genes in *Efm* include those encoding a glycoside hydrolase (*hylEfm*) and several surface proteins involved in adhesion, biofilm formation, and pili assembly, such as *acm*, *scm*, *sgrA*, and *ecbA* (17–20). Additionally, hospital-associated VREfm strains often express more putative virulence markers (PVMs) than community-associated VREfm strains, including the *esp* gene. (17, 21–23). With the onset of WGS, the sequences of increasingly more *Efm* isolates have been analyzed, resulting in the identification of several new PVMs (18, 24–30).

Building on the recently published VirulenceFinder for *Enterococcus faecium* (*Efm*) and *Enterococcus lactis* (*E. lactis*), which hosts a database of 27 virulence genes, we incorporated an additional eight genes associated with bloodstream isolates into the database. Using this enhanced tool, we characterized the virulence profiles of the dominant VREfm clones in Denmark from 2015 to 2023 (31). The seven dominating clones from 2015 to 2022 have been described in detail by Hammerum et al. in 2024 (32).

The primary aim of this study was to characterize the virulence profiles of these dominant VREfm clones to identify clone-specific virulence markers. Additionally, we describe the eight new genes recently added to VirulenceFinder, all of which are linked to bacteremia, providing novel insights into the genetic basis of virulence in clinically significant clones.

## MATERIALS AND METHODS

### Blood isolate collection

In Denmark, clinical VREfm isolates are not notifiable. Since 2015, clinical VREfm isolates have been voluntarily submitted to Statens Serum Institut by the 10 Danish departments of clinical microbiology as part of national surveillance. In this study, we included 516 VREfm blood culture isolates from Danish hospitals between 2015 and 2023. All isolates were collected from unique patients.

### Sequence data, whole-genome sequencing, and sequence analysis

Paired-end sequence data were sequenced at Statens Serum Institut, Denmark. Genomic DNA was extracted (DNeasy blood and tissue kit; Qiagen, Copenhagen, Denmark or MagNA Pure 96 DNA Multi-Sample Kit (Life Technologies, Carlsbad, CA, USA) with subsequent library construction (Nextera kit; Illumina, Little Chesterford, UK) and whole-genome sequencing (MiSeq or NextSeq 550; Illumina, Little Chesterford, UK) according to the manufacturer's instructions, to obtain paired-end reads of 2 × 250 bp or 2 × 150 bp. Raw sequencing data were analyzed using the Bifrost pipeline v. 2.0.8 at Statens Serum Institut (https://github.com/ssi-dk/bifrost) with accepted average coverage of 30× or higher (avg_coverage ≥30). WGS data were either used as raw data

or were *de novo* assembled using SKESA v. 2.2, generated with Bifrost (33). MLST was inferred with the Bifrost pipeline, using the pubMLST database (34). The genomes were clustered using the cgMLST scheme for *E. faecium* by de Been et al. in SeqSphere + v.8.0.1 (Ridom, Münster, Germany), with the suggested threshold of ≤20 allele differences (35). The cgMLST analysis was subsequently clustered by applying single linkage clustering (SLC), with a maximum allelic distance of 20. Local SLC clusters were named after the complex type of the earliest observed *Efm* isolate within each cluster as previously described (32).

## Phylogeny

To compare the results from the *Efm-Elts* virulence database with the distribution of gene variants, we constructed a phylogenetic tree based on single-nucleotide polymorphisms (SNP), using the Northern Arizona SNP Pipeline v. 1.2 (NASP) (36). Duplicated regions of the *Efm* reference chromosome *Efm_DO* with plasmids (Genbank accession: CP003583.1, CP003584.1, CP003585.1, and CP003586.1) were identified using NUCmer (37), by aligning the reference against itself. Draft genomes were mapped against the reference using Burrows-Wheeler Aligner (BWA) (38), and SNPs were identified using GATK (39). Recombinant regions were purged by using Gubbins v. 2.2 (40). The purged alignment was used to reconstruct the phylogeny, performed by maximum-likelihood approximation using the generalized time-reversible model in FastTree v. 2.1.8 (41). The phylogeny was visualized using the Interactive Tree Of Life (iTOL) and annotated with relevant metadata (42).

## The *Efm-Elts* virulence database

The various PVM variants were classified by comparing them to the virulence gene variants, previously identified by Qin et al., which were extracted from hospital-associated (HA) and community-associated (CA) genomes (43). If the *Efm* hospital variant (HV) and community variant (CV) were identical, it was classified as an *Efm* gene variant (V).

The published database consisted of the following genes and gene clusters: *acm, scm, sgrA, ecbA, fnm, sagA, hylEfm, ptsD, orf1481, fms15, fms21*(pilA)-*fms20* (pili gene cluster 1, PGC-1), *fms14-fms17-fms13* (PGC-2), *ebpA-ebpB-ebpC* (PGC-3), *fms11-fms19-fms16* (PGC-4), *ccpA, bepA, gls20-glsB1*, and *gls33-glsB* (General stress proteins, GSP) (31). In this study, the database was expanded with the incorporation of eight new genes, namely *prpA, tirE1, tirE2, capD, lysM1, lysM2, lysM3,* and *lysM4*.

The PVMs in the database are involved in a variety of cellular functions, each of which will be described in more detail. These include (i) cell wall-anchored proteins that play key roles in surface adhesion to the extracellular matrix. This category includes proteins, such as *acm, scm, fnm, sgrA, ecbA*, and *fms15*, the enterococcal surface protein—the *esp* gene variant, and four pilin gene clusters, along with the newly imported *prpA* gene (22, 44, 45). (ii) Carbohydrate metabolism and cell growth, which are facilitated by *orf1481, ccpA*, along with the newly imported TIR-domain containing locus. This locus, which includes *tirE1* and *tirE2*, promotes survival and growth in human blood (24). (iii) Phosphotransferase systems (PTSs) which are critical for carbohydrate transport, including genes such as *ptsD, bepA*. (iv) Extracellularly secreted proteins that may play roles in cell growth, biofilm formation (e.g., *sagA*), colonization loads (e.g., *hylEfm*), and tissue adhesion. The newly imported genes *lysM1, lysM2*, and *lysM3* are part of this group (25). (v) A newly imported gene involved in host colonization and a role in peptidoglycan synthesis (*lysM4*). (vi) General stress proteins that are crucial for bile salt tolerance and intestinal adaptation, such as gls-like proteins (46).

The novel genes used in the updated version of VirulenceFinder will be described in more detail. In a study by Wagner et al., the TIR locus (Toll/interleukin-1 receptor domain), which contains two TIR-domain proteins in *E. faecium* and their role as novel virulence factors is described (24). These proteins play a crucial role in protein-protein interactions within TLR signaling and have been shown to contribute to immune evasion in various bacterial species. Notably, the *tirE* locus is found exclusively in

hospital-associated isolates and resides on a putative mobile genetic element of phage origin, a characteristic shared with many previously described *tir* genes. The study suggests that the *tirE* genes contribute to *E. faecium* pathogenicity, particularly in the context of bloodstream infections.

The LysM domain, a conserved carbohydrate-binding module found in bacterial extracellular proteins, like peptidoglycan hydrolases, adhesins, and virulence factors, plays a significant role in bacterial function and is notably upregulated during infection (25).

In a study by Prieto et al., the surface protein named *prpA* was analyzed for its ability to bind to extracellular matrix components and platelets. The study established *prpA* as a distinctive surface protein in *E. faecium* and related enterococci, characterized by a unique N-terminal domain (47). This N-terminal domain of *prpA* interacts with fibrinogen, fibronectin, and platelets. The temperature-regulated production of *prpA* in clinical *E. faecium* isolates suggests hat it may play a specific, albeit undefined, role in mammalian colonization and infection (47).

The *capD* gene, a polysaccharide protein, has been implicated in biofilm formation. A mouse knockout model showed that disruption of this gene led to reduced growth and reduced biofilm formation (48).

A description of the 35 PVMs, variants along with reference genomes, positions in references, and locus tags is provided in Table S1.

## RESULTS

### The *Efm-Elts* virulence database

Our updated *Efm-Elts* virulence database consists of 35 PVMs. For some of the VREfm isolates, two hits to the *ccpA* gene were reported. When assessing the *ccpA* community variant control strain C59, the results also showed two hits. The *Efm* hospital gene variant showed 97.45% identity and 99.50% coverage with a depth of 18.22× compared with the correct *Efm* community gene variant, which had 100% identity and 100% coverage with a depth of 231.29×. Additionally, when assessing the other PVM gene hits for C59, the depth was a minimum of 114×, indicating that the extra *ccpA* hit may be attributed to sequencing error rather than genuine genetic variation. The same pattern was observed with the VREfm isolates, and therefore only the best hit with high depth was reported.

Most clonal groups exhibited a high prevalence (≥80%) of key genes associated with surface-exposed, cell-wall-anchored proteins, including *acm, scm, fnm, sgrA,* and *fms15*. Additionally, all clonal groups contained the three virulence genes—*empA, empB,* and *empC*—from the PGC-3 in 100% of the isolates. The *fms13* gene from the PGC-2 was also present in all clonal groups (Fig. 1A).

Some variation was observed in the genes associated with carbohydrate metabolism and cell growth, with *ccpA* being present in all isolates across the clonal groups, though in different variants (Fig. 1A).

Regarding genes related to biofilm formation, adhesion, and colonization, some more pronounced differences were noted between the clonal groups. The differences between the clonal groups were analyzed and are summarized below and described as a heat map in Fig. 1A. The clonal groups are described in order by years when they first emerged and dominated in this collection (Fig. 1B) and presented with a phylogenetic tree in Fig. 2.

In the ST203-CT859 van*A Efm* group, which includes 84 isolates, the putative cell-wall-anchored protein genes *fms20* and *fms21* from the PGC-1 were entirely absent. The gene *ccpA*, associated with carbohydrate metabolism and cell growth, was found in 100% of isolates from this clonal group, but unlike other clonal groups, it was also found in *Elts* isolates. The *hylEfm* gene, which encodes glycosyl hydrolase, was completely absent in this group, while the *lysM3* complex was present in 20-50% of the isolates.

In the ST80-CT14 van*A Efm* group, consisting of 30 isolates, the gene for the collagen-binding protein A (*ecbA*) was completely absent. However, the putative cell-wall anchored protein genes *fms20* and *fms21* from the PGC-1 were present in 80% or more

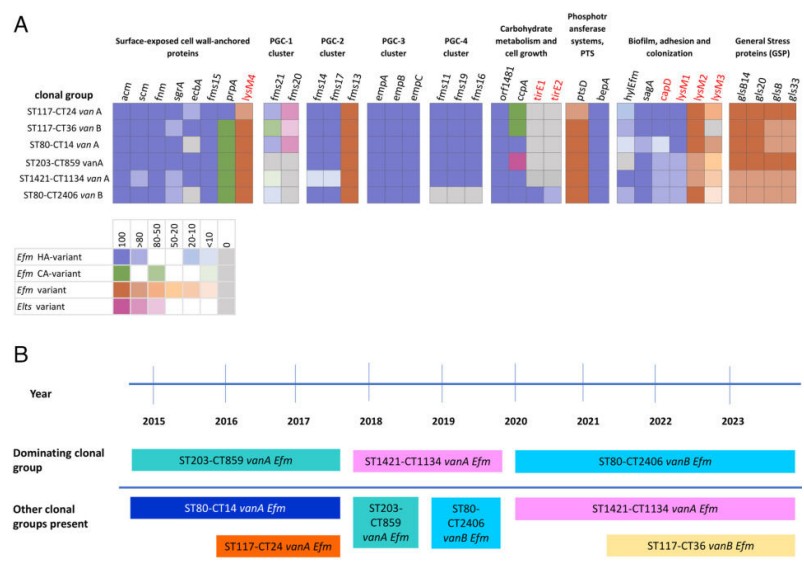

**FIG 1** (A) Heat map of the relationship between the dominant VREfm clonal groups and the 35 PVMs currently part of VirulenceFinder. The red genes represent the eight new genes from the updated version of VirulenceFinder. The color bar at the bottom of the figure illustrates the variant: *Efm*-HV, *E. faecium* hospital variant; *Efm*-CV, *E. faecium* community-variant Efm-V, *E. faecium* variant; Elts-V, *E. lactis* variant. (B) Timeline of the dominant VREfm clone related to year as well as other clonal groups present.

of the isolates. The *hylEfm* gene was present in less than 10% of the isolates, and the *capD* gene was found in less than 10% of the isolates in this clonal group, whereas it was present in 80-100% of isolates in all other clonal groups. The *lysM3* complex was present in more than 80% of the isolates from this group.

In the ST117-CT24 van*A Efm* group, which includes 20 isolates, the putative cell-wall-anchored protein genes *fms20* and *fms21* from the PGC-1 were present in 80% or more of the isolates. Genes associated with carbohydrate metabolism and cell growth, *orf1481* and *ccpA*, were found in 100% of the isolates from this clonal group, with *orf1481* being present in all community-variant isolates. The *hylEfm* gene was present in less than 20% of the isolates, while the *lysM3* complex was present in 50%–80% of the isolates.

In the ST1421-CT1134 van*A Efm* group, which includes 104 isolates, the putative cell-wall-anchored protein genes *fms20* and *fms21* from the PGC-1 were entirely absent (*fms20*) or almost entirely absent (*fms21*). From the PGC-2, fewer than 10% of the isolates carried the *fms14* and *fms17* genes, whereas these genes were present in 100% of isolates from other clonal groups. The *hylEfm* gene was present in 100% of the isolates from this clonal group, while the lysM3 complex was present in less than 20% of the isolates.

In the ST80-CT2406 van*B Efm* group, which includes 140 isolates, the gene for the collagen-binding protein A (*ecbA*) was completely absent. The putative cell-wall-anchored protein gene *fms20* from the PGC-1 was also completely absent, while *fms21* was found in less than 10% of the isolates. Additionally, all genes from the PGC-4 (*fms11*, *fms19*, and *fms16*) were completely absent in this group, while they were present in 100% of the isolates from other clonal groups. The *tirE* complex, which encodes the *tirE1* and *tirE2* genes, was present in more than 80% of the isolates from this clonal group, whereas it was either completely absent or present in fewer than 10% of the isolates from other clonal groups. The *hylEfm* gene was present in more than 80% of the isolates, while the lysM3 complex was present in less than 10% of the isolates.

In the ST117-CT36 van*B Efm* group, which includes 21 isolates, the putative cell-wall-anchored protein genes *fms20* and *fms21* from the PGC-1 were present in 50%–80% of the isolates. The *hylEfm* gene was present in more than 80% of the isolates, while the *lysM3* complex was completely absent in this clonal group.

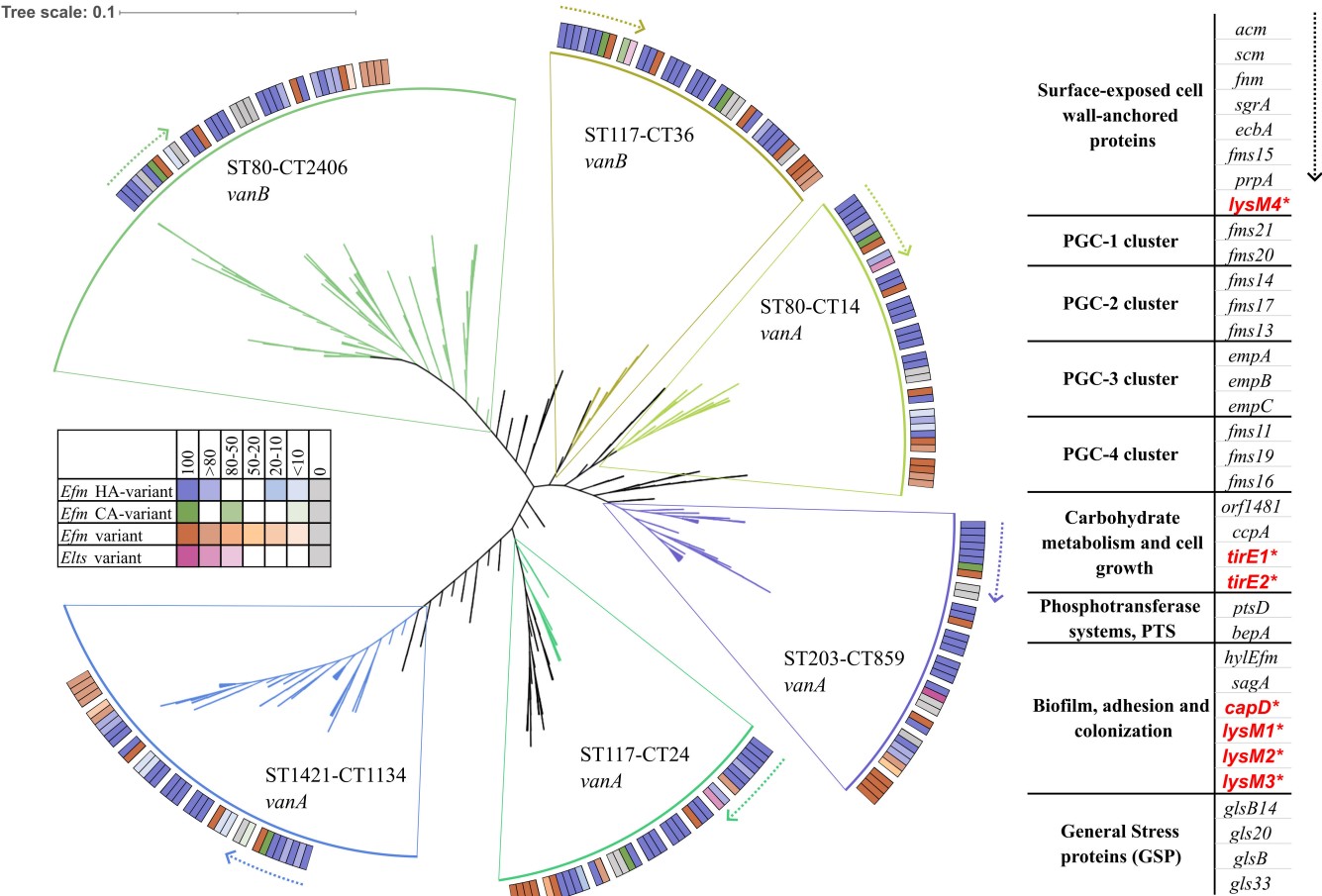

**FIG 2** Phylogeny of the six dominant VREfm clusters with a heatmap of the 35 PVMs. Genes newly included in the updated VirulenceFinder database are highlighted in bold red and marked with an asterisk *.

## DISCUSSION

Since 2012, Denmark has experienced a rise in VREfm infections due to the emergence of multiple *vanA* VREfm clones (49). However, from 2019 onwards, *vanB* VREfm began to increase and eventually outcompeted the *vanA* strains. (49, 50). As observed in many other countries, Denmark has witnessed a continuous clonal shift among endemic VREfm clones (32).

Enterococci are known for their remarkable resistance to various environmental stresses. Research has shown that outbreak-related strains possess specific gene repertoires typically absent in commensal or environmental strains. These genes likely enable enterococci to thrive in harsh hospital settings (51). While the microbiota of a healthy individual contains *E. faecium* in the order of 0.1%, in the dysbiotic microbiota of hospitalized patients, *E. faecium* becomes the dominant bacterial species (52, 53). The acquisition and enhancement of specific genes encoding carbohydrate uptake systems in VREfm strains from hospital settings may equip them with a broader or novel metabolic repertoire, enabling efficient colonization of the dysbiotic intestinal microbiota of hospitalized patients.

Our group previously reported that the clustering of our VREfm blood culture collection closely mirrors the clustering of a larger collection of clinical VREfm isolates from Denmark (32). From that study, we learned that from 2015 to 2022, several *E. faecium* clones spread across Denmark. ST117-CT24 *vanA* spread nationwide, while ST117-CT36 *vanB Efm* emerged in the Capital Region in 2019 and quickly spread across all regions. ST80-CT14 *vanA Efm* was widespread in 2015 but declined by 2022, with

only a few isolates remaining in the Capital Region. ST203-CT859 *vanA Efm* dominated from 2015 to 2018, then declined. ST1421-CT1134 *vanA Efm* became prevalent in 2018, peaked in 2019, and then declined. Since 2019, ST80-CT2406 *vanB Efm* has been the most dominant clonal type in Denmark.

An overview of the dominant clusters revealed a variable pattern of PVMs. ST117-CT24 *vanA* showed robust presence across nearly all genes, indicating high adaptability and virulence potential. Markedly, it did not contain the *hylEfm* gene, which indeed has been inconsistently detected across different epidemic clones (17). Likewise, ST117-CT36 *vanB Efm* generally showed high percentages of all genes. For ST80-CT14 *vanA Efm,* there were a few notable absences in the group encoding for biofilm, adhesion, and colonization, possibly explaining its niche being taken up by a more fit clone. ST203-CT859 *vanA Efm* displayed a diverse profile with some genes absent, particularly in the PGC-1, indicating potential weaknesses in colonization or nutrient utilization. ST1421-CT1134 *vanA Efm* showed variability in the presence of certain genes, particularly in the PGC-1 and PGC-2, suggesting functional diversity. ST80-CT2406 *vanB Efm* exhibited a similar pattern with high presence across most genes, and the most notable absences to be found in the PGC-1 and PGC-4.

Compared with *Elts* and *Efm* CA variants, HA-associated *Efm* isolates frequently possess PVMs, including the *acm* gene (which encodes a collagen-binding adhesin), the *ebpABC* operon that is critical for adherence and biofilm formation, *hylEfm* that is a glycoside hydrolase, and most of the 14 *fms* genes that encode *E. faecium* surface proteins, including pili (46). Some specific genes that encode colonization and biofilm formation are *sagA* and *hylEfm. sagA* is the only virulence factor known to be specific to *Efm* and the hospital-associated variant was present at 100% in all our six clonal groups. *hylEfm* does not affect *E. faecium* colonization *per se* but isolates with the *hylEfm* gene on plasmids show higher bacterial loads. In fact, *hylEfm*-carrying mega plasmids that enhance bacterial proliferation in blood survival assays carry other potential virulence factors, such as genes involved in metabolism and cell growth (54, 55). In our collection, *hylEfm* was present at 80% or above in ST117-CT36 *vanB Efm*, ST1421-CT1134 *vanA Efm,* and ST80-CT2406 *vanB Efm*. Two of these groups contained some of the most successful CT clones, namely CT1134 *vanA Efm* and CT2406 *vanA Efm*. With their role in biofilm formation, which is critical for persistence in host environments, *hylEfm* and *sagA* could be part of the explanation for how these clones have been so successful at establishing themselves in the hospital environment.

Genes from the group of surface-exposed cell-wall-anchored proteins are conserved among all *Efm* isolates and play crucial roles in bacterial cell wall structure and adhesion to the extracellular matrix. From a previous study, we noted that all our isolates from the hospital-associated group contained these genes (*acm*, *scm*, *fnm*, *sgrA,* and *fms15*)(31).

From the PGC-3, the genes in this group have been shown as important markers of *Efm* to cause urinary tract infection (56). Two of the genes, *empA* and *empB,* are also important for biofilm and adherence to extracellular matrix proteins.

Genes from the PGC-1 have been linked to clinical *Efm* isolates. Between 2005 and 2010, several studies found that specific LPXTG surface proteins and PGCs, such as *esp*, *sgrA* (*orf2351*), and *ecbA* (*orf2430*), were more common clinical and outbreak-associated *Efm* clonal complex 17 isolates. (19, 27, 57).

The updated version of VirulenceFinder contains eight genes in the group of surface-anchored proteins, namely, *acm, scm, fnm, sgrA, ecbA, fms15, prpA,* and *lysM4*. These are all important putative virulence determinants. In a longitudinal study, it was shown that among clinical outbreak ST78 *E. faecium* isolates, the prevalence of many of these virulence factors (*acm, scm, sgrA, ecbA* as well as *esp* and *pilA* and *pilB*) had a significantly higher prevalence (58). The only noticeable difference in our collection was that *ecbA* was completely absent in ST80-CT2406 *vanB Efm* and ST80-CT14 *vanA Efm*. All other groups contained all the tested PVMs from the group of surface-anchored proteins in high numbers (between 80% and 100% of isolates). Surface proteins are integral to bacterial interactions with their environment. They facilitate host cell adhesion and

invasion, detect environmental conditions, transmit signals to the cytoplasm, and help mount defenses against host responses. Two peptidoglycan-associated proteins, *sagA* and *lysM4*, implicated in virulence have been evaluated for their potential as vaccine candidates against *E. faecium* (59). Thus, it is not surprising that all our dominant groups contained these genes. In our study, *prpA* was identified in all dominant clones, predominantly within the CA-*Efm* variants.

For the two PVMs (*ptsD* and *bepA*) in the group of phosphotransferase system (PTS), all clonal groups contained them in 100% of all isolates. PTS are transmembrane proteins that phosphorylate and transport specific carbohydrates across the bacterial cell membrane, and they are thought to be involved in enterococcal stress response (51).

Another shared trait among all our clonal groups was that they all contained general stress proteins in 80%–100% of all isolates. *glsB, gls20, and gls33* have all been associated with resistance to bile salts (46).

Of particular interest, our analysis revealed that only the latest emerging clone, namely ST80-CT2406 *vanB Efm,* carried the *tirE* genes, where this could possibly be an emerging trait contributing to its success. This group, by far the largest clonal cluster in our collection, included 140 isolates out of a total of 516 blood isolates. As mentioned previously, the *tirE* genes are linked to enhanced bacterial proliferation and survival in blood, which may explain the predominance of this group among the blood isolates (24).

## Conclusion

By analyzing the updated version of the recently released VirulenceFinder database for *E. faecium* and *E. lactis*, we identified distinct patterns of PVMs across dominant VREfm clusters, particularly those linked to biofilm formation, metabolism, carbohydrate metabolism, and survival and proliferation in blood. These patterns may help explain variations in pathogenicity, metabolic capabilities, and stress adaptation, potentially contributing to the shift in endemic clones, which is a defining characteristic of VREfm.

## ACKNOWLEDGMENTS

Frank Hansen, Pia Thurø Hansen, and Line Toft Madsen at Statens Serum Institut are thanked for their excellent laboratory assistance.

## AUTHOR AFFILIATIONS

[1]Department of Bacteria, Parasites and Fungi, Statens Serum Institut, Copenhagen, Denmark

[2]Department of Virus & Microbiological Special Diagnostics, Statens Serum Institut, Copenhagen, Denmark

[3]Department of Clinical Microbiology, Copenhagen University Hospital-Rigshospitalet, Copenhagen, Denmark

[4]Research Group for Host-Microbe Interactions, Department of Medical Biology, Faculty of Health Sciences, UiT, The Arctic University of Norway, Tromsø, Norway

[5]Norwegian Centre for Detection of Antimicrobial Resistance, Department of Microbiology and Infection Control, University Hospital of North Norway, Tromsø, Norway

[6]UCIBIO, Unidade de Ciências Biomoleculares Aplicadas, Faculdade de Farmácia, Universidade do Porto, Porto, Portugal

[7]Laboratório Associado i4HB Instituto para a Saúde e a Bioeconomia, Faculdade de Farmácia, Universidade do Porto, Porto, Portugal

[8]UCIBIO, Unidade de Ciências Biomoleculares Aplicadas, Instituto Universitário de Ciências da Saúde (1H-TOXRUN, IUCS-CESPU), Gandra, Portugal

## AUTHOR ORCIDs

Ingrid Maria Cecilia Rubin  http://orcid.org/0000-0002-6158-2927

Kasper Thystrup Karstensen ⓘ https://orcid.org/0000-0002-5009-0476
Kristin Hegstad ⓘ https://orcid.org/0000-0002-1314-0497
Ana R. Freitas ⓘ http://orcid.org/0000-0002-7326-4133
Anette M. Hammerum ⓘ http://orcid.org/0000-0001-9471-1601
Louise Roer ⓘ http://orcid.org/0000-0001-6892-913X

## AUTHOR CONTRIBUTIONS

Ingrid Maria Cecilia Rubin, Conceptualization, Formal analysis, Investigation, Methodology, Project administration, Visualization | Kasper Thystrup Karstensen, Conceptualization, Data curation, Formal analysis, Methodology, Validation | Mikkel Lindegaard, Conceptualization, Methodology, Visualization | Kristin Hegstad, Investigation, Supervision | Ana R. Freitas, Validation | Anette M. Hammerum, Conceptualization, Data curation, Project administration, Resources, Supervision | Louise Roer, Conceptualization, Data curation, Formal analysis, Investigation, Project administration, Software, Supervision, Visualization

## DATA AVAILABILITY

Accession numbers for the Illumina short-reads sequences included in this study can be found in Table S2.

## ETHICS APPROVAL

This article has been prepared on the basis of a study carried out as part of a task imposed on Statens Serum Institut according to the national legislation as specified in the Danish Health Care Act (Sundhedsloven § 222). The need for ethics approval and informed consent is therefore deemed unnecessary according to national legislation, cf. implementing decree 2020-09-01, number 1338, about scientific regulatory procedure of health science research projects and health data scientific research projects. The article only contains aggregated results and no personal data. The article is therefore not covered by the European General Data Protection Regulation.

## ADDITIONAL FILES

The following material is available online.

### Supplemental Material

**Table S1 (Spectrum01289-25 s0001.xlsx).** Gene content.
**Table S2 (Spectrum01289-25 s0002.xlsx).** Accession numbers.

### Open Peer Review

**PEER REVIEW HISTORY (review-history.pdf).** An accounting of the reviewer comments and feedback.

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
