## [Reviewer comments · Microbiology Spectrum]

Microbiology Spectrum

Virulence signature of the endemic Vancomycin-Resistant *Enterococcus faecium* clones in Denmark, 2015-2023

Ingrid Rubin, Kasper Karstensen, Mikkel Lindegaard, Kristin Hegstad, Ana Freitas, Anette Hammerum, and Louise Roer

Corresponding Author(s): Louise Roer, Statens Serum Institut Bakterier Parasitter & Svampe

Review Timeline:

Submission Date:	April 24, 2025
Editorial Decision:	June 16, 2025
Revision Received:	June 19, 2025
Accepted:	June 27, 2025

Editor: Vittal Ponraj

Reviewer(s): The reviewers have opted to remain anonymous.

Transaction Report:

DOI: <https://doi.org/10.1128/spectrum.01289-25>

Re: Spectrum01289-25 (Virulence signature of the endemic Vancomycin-Resistant Enterococcus faecium clones in Denmark, 2015-2023)

Dear Dr. Louise Roer:

Thank you for the privilege of reviewing your work. Below you will find my comments, instructions from the Spectrum editorial office, and the reviewer comments.

Revision Guidelines

Sincerely,

Vittal Prakash Ponraj Ph.D SM(ASCP)CM
Editor
Microbiology Spectrum

Reviewer #1 (Public repository details (Required)):

sequence data in SRA/NCBI.

Reviewer #1 (Comments for the Author):

Rubin et al., wrote a very relevant, very interesting, concise and easy to read paper about the rise of vancomycin-resistant

Enterococcus faecium (VREfm) in Denmark since 2012 which is linked to the species' genetic adaptability, with new virulence genes and dominant clones identified, highlighting evolving pathogenicity and resistance patterns. This enabled the group to update VirulenceFinder for *E. faecium*.

One major issue with this manuscript is that the result section from particularly lines 196-240 is written and reads more like "bullet points" or statements about the findings rather than in a fluent article like style. The authors are encouraged to make this section more fluent.

The following minor issues should be addressed prior publication in ASM Microbiology Spectrum:

Line 27: virulence markers (PVMs). Does PVM mean "putative virulence markers"?

Line 30: probably putative virulence markers? Since no animal models were used to test potential virulence defects in PVM knockout vs WT strains?

Line 41: *E. faecium* and *E. lactis* should be *E. faecium* and *E. lactis*

Line 55: van clusters should be van clusters

Line 62: please use either clade or Clade consistently throughout.

Lines 70-73: Please cite the original work where possible. Acm is described in a beautiful Mol. Micro article of Nallapareddy et al., Clinical isolates of *Enterococcus faecium* exhibit strain-specific collagen binding mediated by Acm, a new member of the MSCRAMM family - PubMed

Scm by Sillanpaa et al.: Identification and phenotypic characterization of a second collagen adhesin, Scm, and genome-based identification and analysis of 13 other predicted MSCRAMMs, including four distinct pilus loci, in *Enterococcus faecium* - PubMed

SgrA and EcbA by Hendrickx et al.: SgrA, a nidogen-binding LPXTG surface adhesin implicated in biofilm formation, and EcbA, a collagen binding MSCRAMM, are two novel adhesins of hospital-acquired *Enterococcus faecium* - PubMed and later: The crystal structure of the ligand-binding region of serine-glutamate repeat containing protein A (SgrA) of *Enterococcus faecium* reveals a new protein fold: functional characterization and insights into its adhesion function - PubMed

Enterococcal surface protein Esp should be mentioned here as well: Variant esp gene as a marker of a distinct genetic lineage of vancomycin-resistant *Enterococcus faecium* spreading in hospitals - PubMed

Or reviewed for the first time by Hendrickx et al., LPxTG surface proteins of enterococci - PubMed and The cell wall architecture of *Enterococcus faecium*: from resistance to pathogenesis - PubMed

Lines 73-74: putative virulence markers (PVMs) is also described in Line 27.

Line 137: consider citing LPxTG surface proteins of enterococci - PubMed or The cell wall architecture of *Enterococcus faecium*: from resistance to pathogenesis - PubMed

Line 138: pili gene clusters should be pilin gene cluster, or PGC, first mentioned in Pila/PilB *E. faecium* pili paper: Expression of two distinct types of pili by a hospital-acquired *Enterococcus faecium* isolate - PubMed

Lines 137-138: *Enterococcal surface protein Esp* is missing here?

Line 175: For the ccpA, gene, should be For the ccpA gene.

Line 178: 97.45% ID and 99.50% Cov, ID= identity and Cov= coverage? Please write full when used for the first time

Line 272-272: the PGC-1 and PGC-2 clusters should be: the PGC-1 and PGC-2, since PGC= pilin gene cluster. Adding clusters would be double.

Line 294: the reference in this sentence seems incorrect? Multiple studies in the years 2005-2010 demonstrated that particular LPXTG type of surface proteins and PGCs were enriched in hospital adapted strains, including for Esp, SgrA (orf2351), EcbA (orf2430), PGCs: Five genes encoding surface-exposed LPXTG proteins are enriched in hospital-adapted *Enterococcus faecium* clonal complex 17 isolates - PubMed, SgrA, a nidogen-binding LPXTG surface adhesin implicated in biofilm formation, and EcbA, a collagen binding MSCRAMM, are two novel adhesins of hospital-acquired *Enterococcus faecium* - PubMed and Expression of two distinct types of pili by a hospital-acquired *Enterococcus faecium* isolate - PubMed

Line 300: These are all important virulence determinants. I wonder whether this is really true for all the MSCRAMMs. Are all mentioned MSCRAMMs here, WT and knockout MRSAMMs tested in animal models? Probably, adding "putative" would be more on the safe side: These are all important putative virulence determinants.

Figures: "PGC-1 cluster" means "Pilin gene cluster-1 cluster", so adding "cluster" after PGC is not necessary. Please change throughout.

Figure 1b: please align the year numbers on a straight line.

Figure legend: please write bacterial species in italics.

References: please write bacterial species and gene names in italics.

Reviewer #2 (Comments for the Author):

This report from Rubin and colleagues describes sequencing and genomic analysis of a large panel of vancomycin-resistant *Enterococcus faecium* (VREfm) isolates from Denmark. VREfm is a notable public health concern. The authors use this data to identify genomic features and putative virulence factors of several Efm sequence types and clusters over time. This data builds upon the previous release of a virulence factor database for Efm. This paper describes 6 dominant VREfm clusters and potential virulence factors distributed throughout these clusters, including the tir genes which promote growth in blood. Overall, this is a concise description of a major genomic effort focused on identifying potential virulence factors in a major opportunistic pathogen. The work should be broadly relevant for other *Enterococcus* researchers as well as other epidemiologists interested in the genomics of Gram-positive pathogens. I have a few minor comments regarding the clarity of the manuscript:

- 1) It seems that PVM is a bit unnecessary as an acronym given that the manuscript has a fair amount of other acronyms and abbreviations. I suggest spelling out PVM, as this would not significantly increase the word count but would help with clarity for the reader.
- 2) Does the collection need some kind of study or approval number in the methods section?
- 3) Lines 177-180: the rationale for strain C59 is not well described. Any data mentioned should be shown.
- 4) starting at line 196: these sections would flow better if the sentences were combined into a paragraph, but perhaps that is something that could be corrected during copy editing.
- 5) Figure 2: the red/black indications will be lost if the paper is viewed in grayscale. For accessibility, I suggest using bold text instead of red, or adding another column to signify the newly imported genes.

Rubin et al., wrote a very relevant, very interesting, concise and easy to read paper about the rise of vancomycin-resistant *Enterococcus faecium* (VREfm) in Denmark since 2012 which is linked to the species' genetic adaptability, with new virulence genes and dominant clones identified, highlighting evolving pathogenicity and resistance patterns. This enabled the group to update VirulenceFinder for *E. faecium*.

One major issue with this manuscript is that the result section from particularly lines 196-240 is written and reads more like “bullet points” or statements about the findings rather than in a fluent article like style. The authors are encouraged to make this section more fluent.

The following minor issues should be addressed prior publication in ASM Microbiology Spectrum:

Line 27: virulence markers (PVMs). Does PVM mean “putative virulence markers”?

Line 30: probably putative virulence markers? Since no animal models were used to test potential virulence defects in PVM knockout vs WT strains?

Line 41: *E. faecium* and *E. lactis* should be *E. faecium* and *E. lactis*

Line 55: van clusters should be *van* clusters

Line 62: please use either clade or Clade consistently throughout.

Lines 70-73: Please cite the original work where possible. Acm is described in a beautiful Mol. Micro article of Nallapareddy et al., Clinical isolates of Enterococcus faecium exhibit strain-specific collagen binding mediated by Acm, a new member of the MSCRAMM family - PubMed

Scm by Sillanpaa et al.,: Identification and phenotypic characterization of a second collagen adhesin, Scm, and genome-based identification and analysis of 13 other predicted MSCRAMMs, including four distinct pilus loci, in Enterococcus faecium - PubMed

SgrA and EcbA by Hendrickx et al.,: SgrA, a nidogen-binding LPXTG surface adhesin implicated in biofilm formation, and EcbA, a collagen binding MSCRAMM, are two novel adhesins of hospital-acquired Enterococcus faecium - PubMed and later: The crystal structure of the ligand-binding region of serine-glutamate repeat containing protein A (SgrA) of Enterococcus faecium reveals a new protein fold: functional characterization and insights into its adhesion function - PubMed

Enterococcal surface protein Esp should be mentioned here as well: Variant esp gene as a marker of a distinct genetic lineage of vancomycin-resistant Enterococcus faecium spreading in hospitals - PubMed

Or reviewed for the first time by Hendrickx et al., LPxTG surface proteins of enterococci - PubMed and The cell wall architecture of Enterococcus faecium: from resistance to pathogenesis - PubMed

Lines 73-74: putative virulence markers (PVMs) is also described in Line 27.

Line 137: consider citing LPxTG surface proteins of enterococci - PubMed or The cell wall architecture of Enterococcus faecium: from resistance to pathogenesis - PubMed

Line 138: pili gene clusters should be pilin gene cluster, or PGC, first mentioned in Pila/PilB *E. faecium* pili paper: Expression of two distinct types of pili by a hospital-acquired Enterococcus faecium isolate - PubMed

Lines 137-138: Enterococcal surface protein Esp is missing here?

Line 175: For the *ccpA*, gene, should be For the *ccpA* gene.

Line 178: 97.45% ID and 99.50% Cov, ID= identity and Cov= coverage? Please write full when used for the first time

Line 272-272: the PGC-1 and PGC-2 clusters should be: the PGC-1 and PGC-2, since PGC= pilin gene cluster. Adding clusters would be double.

Line 294: the reference in this sentence seems incorrect? Multiple studies in the years 2005-2010 demonstrated that particular LPXTG type of surface proteins and PGCs were enriched in hospital adapted strains, including for Esp, SgrA (orf2351), EcbA (orf2430), PGCs: Five genes encoding surface-exposed LPXTG proteins are enriched in hospital-adapted Enterococcus faecium clonal complex 17 isolates - PubMed, SgrA, a nidogen-binding LPXTG surface adhesin implicated in biofilm formation, and EcbA, a collagen binding MSCRAMM, are two novel adhesins of hospital-acquired Enterococcus faecium - PubMed and Expression of two distinct types of pili by a hospital-acquired Enterococcus faecium isolate - PubMed

Line 300: These are all important virulence determinants. I wonder whether this is really true for all the MSCRAMMs. Are all mentioned MSCRAMMs here, WT and knockout MRSAMMs tested in animal models? Probably, adding “putative” would be more on the safe side: These are all important putative virulence determinants.

Figures: “PGC-1 cluster” means “Pilin gene cluster-1 cluster”, so adding “cluster” after PGC is not necessary. Please change throughout.

Figure 1b: please align the year numbers on a straight line.

Figure legend: please write bacterial species in italics.

References: please write bacterial species and gene names in italics.

Please find below our point-by point response to the reviewers:

Reviewer #1 (Comments for the Author):

Rubin et al., wrote a very relevant, very interesting, concise and easy to read paper about the rise of vancomycin-resistant *Enterococcus faecium* (VREfm) in Denmark since 2012 which is linked to the species' genetic adaptability, with new virulence genes and dominant clones identified, highlighting evolving pathogenicity and resistance patterns. This enabled the group to update VirulenceFinder for *E. faecium*.

Thank you for your kind and encouraging words regarding our manuscript. We are pleased to hear that you found our paper on the virulence profile of vancomycin-resistant *Enterococcus faecium* (VREfm) in Denmark since 2012 to be relevant.

One major issue with this manuscript is that the result section from particularly lines 196-240 is written and reads more like "bullet points" or statements about the findings rather than in a fluent article like style. The authors are encouraged to make this section more fluent.

Response: Thank you, that is a good point. The section has now been rewritten in a fluent article like style. Please see lines 196-238, which have been highlighted in the marked-up file.

The following minor issues should be addressed prior publication in ASM Microbiology Spectrum:

Line 27: virulence markers (PVMs). Does PVM mean "putative virulence markers"?

Response: that is correct and has been changed

Line 30: probably putative virulence markers? Since no animal models were used to test potential virulence defects in PVM knockout vs WT strains?

Response: Correct, putative has been added to line 31

Line 41: *E. faecium* and *E. lactis* should be *E. faecium* and *E. lactis*

Response: Thank you, this has been changed

Line 55: van clusters should be van clusters

Response. I believe you mean it should be italicized, which is now done.

Line 62: please use either clade or Clade consistently throughout.

Response: thank you. This has been changed and highlighted throughout

Lines 70-73: Please cite the original work where possible. *Acm* is described in a beautiful Mol. Micro article of Nallapareddy et al., Clinical isolates of *Enterococcus faecium* exhibit

strain-specific collagen binding mediated by Acm, a new member of the MSCRAMM family - PubMed

Scm by Sillanpaa et al.,: Identification and phenotypic characterization of a second collagen adhesin, Scm, and genome-based identification and analysis of 13 other predicted MSCRAMMs, including four distinct pilus loci, in *Enterococcus faecium* - PubMed

SgrA and EcbA by Hendrickx et al.,: SgrA, a nidogen-binding LPXTG surface adhesin implicated in biofilm formation, and EcbA, a collagen binding MSCRAMM, are two novel adhesins of hospital-acquired *Enterococcus faecium* - PubMed and later: The crystal structure of the ligand-binding region of serine-glutamate repeat containing protein A (SgrA) of *Enterococcus faecium* reveals a new protein fold: functional characterization and insights into its adhesion function - PubMed

Enterococcal surface protein Esp should be mentioned here as well: Variant esp gene as a marker of a distinct genetic lineage of vancomycin-resistant *Enterococcus faecium* spreading in hospitals - PubMed

Or reviewed for the first time by Hendrickx et al., LPxTG surface proteins of enterococci - PubMed and The cell wall architecture of *Enterococcus faecium*: from resistance to pathogenesis – PubMed

Response: Thank you for these excellent references, that have indeed added substantially to our paper. We have added a line about the *esp* variant, please see line 139.

Lines 73-74: putative virulence markers (PVMs) is also described in Line 27.

Response: Yes, correct. That is however in the abstract, consequently we thought it good to mention it also in the introduction, as it is important to the paper.

Line 137: consider citing LPxTG surface proteins of enterococci - PubMed or The cell wall architecture of *Enterococcus faecium*: from resistance to pathogenesis – PubMed

Response. Thank you, this reference has been added.

Line 138: pili gene clusters should be pilin gene cluster, or PGC, first mentioned in Pila/PilB *E. faecium* pili paper: Expression of two distinct types of pili by a hospital-acquired *Enterococcus faecium* isolate – PubMed

Response: Thank you: this reference has been added along with the one above.

Lines 137-138: Enterococcal surface protein Esp is missing here?

Response: The esp has been added along with its reference. Also some additional references have been added to this section. Please see line 139.

Line 175: For the *ccpA*, gene, should be For the *ccpA* gene.

Response: this has been changed

Line 178: 97.45% ID and 99.50% Cov, ID= identity and Cov= coverage? Please write full when used for the first time

Response: this has been corrected

Line 272-272: the PGC-1 and PGC-2 clusters should be: the PGC-1 and PGC-2, since PGC= pilin gene cluster. Adding clusters would be double.

Response: thank you, that is a valid point and it has been changed (lines 270-271)

Line 294: the reference in this sentence seems incorrect? Multiple studies in the years 2005-2010 demonstrated that particular LPXTG type of surface proteins and PGCs were enriched in hospital adapted strains, including for Esp, SgrA (orf2351), EcbA (orf2430), PGCs: Five genes encoding surface-exposed LPXTG proteins are enriched in hospital-adapted Enterococcus faecium clonal complex 17 isolates - PubMed, SgrA, a nidogen-binding LPXTG surface adhesin implicated in biofilm formation, and EcbA, a collagen binding MSCRAMM, are two novel adhesins of hospital-acquired Enterococcus faecium - PubMed and Expression of two distinct types of pili by a hospital-acquired Enterococcus faecium isolate – PubMed

Response: Lines 296-299 have been rewritten with these new references added and a line about it. Thank you

Line 300: These are all important virulence determinants. I wonder whether this is really true for all the MSCRAMMs. Are all mentioned MSCRAMMs here, WT and knockout MRSAMMs tested in animal models? Probably, adding "putative" would be more on the safe side: These are all important putative virulence determinants.

Response. We agree, and putative has been added before virulence determinants on line 302

Figures: "PGC-1 cluster" means "Pilin gene cluster-1 cluster", so adding "cluster" after PGC is not necessary. Please change throughout.

Response: thank you for pointing it out. The extra cluster has been deleted throughout.

Figure 1b: please align the year numbers on a straight line.

Response: This has been done.

Figure legend: please write bacterial species in italics.

Response: This has been done.

References: please write bacterial species and gene names in italics.

Response: bacterial spp. are now in italics in the reference section

Reviewer #2 (Comments for the Author):

This report from Rubin and colleagues describes sequencing and genomic analysis of a large panel of vancomycin-resistant *Enterococcus faecium* (VREfm) isolates from Denmark. VREfm is a notable public health concern. The authors use this data to identify genomic features and putative virulence factors of several Efm sequence types and clusters over time. This data builds upon the previous release of a virulence factor database for Efm. This paper describes 6 dominant VREfm clusters and potential virulence factors distributed throughout these clusters, including the *tir* genes which promote growth in blood. Overall, this is a concise description of a major genomic effort focused on identifying potential virulence factors in a major opportunistic pathogen. The work should be broadly relevant for other *Enterococcus* researchers as well as other epidemiologists interested in the genomics of Gram-positive pathogens. I have a few minor comments regarding the clarity of the manuscript:

Thank you for your thoughtful and detailed review of our manuscript. We appreciate your recognition of the significance of our genomic analysis of vancomycin-resistant *Enterococcus faecium* (VREfm) isolates from Denmark. Your comments on the identification of genomic features and putative virulence factors, as well as the relevance of our work to the broader research community, are highly valued. We are grateful for and have carefully considered your minor comments to improve the clarity of our manuscript.

1) It seems that PVM is a bit unnecessary as an acronym given that the manuscript has a fair amount of other acronyms and abbreviations. I suggest spelling out PVM, as this would not significantly increase the word count but would help with clarity for the reader.

Response: PVM has been used 17 times in the manuscript and is also used in the figures. We would therefore like to keep the acronym as we believe the text gets a bit heavy, should we write it out each time.

2) Does the collection need some kind of study or approval number in the methods section?

Response: The VREfm is part of the national surveillance program in Denmark, thus under the national legislation in the Danish Health Care Act. An Ethics paragraph has been added to line 335-342 in the marked.

3) Lines 177-180: the rationale for strain C59 is not well described. Any data mentioned should be shown.

The C59 was the database control strain for *ccpA*, and therefore assessed for the same pattern. Line 177-185 has been updated to clarify this.

4) starting at line 196: these sections would flow better if the sentences were combined into a paragraph, but perhaps that is something that could be corrected during copy editing.

Response: thank you for a very valid observation. This has also been mentioned by Reviewer 1, and it has now been changed into a more fluent article-style paragraph.

5) Figure 2: the red/black indications will be lost if the paper is viewed in grayscale. For accessibility, I suggest using bold text instead of red, or adding another column to signify the newly imported genes.

Response: thank you for this very important comment. We have updated the figure. New genes are now marked in red, with * and using bold text. This is updated in the figure legend in line 554-556.

Re: Spectrum01289-25R1 (Virulence signature of the endemic Vancomycin-Resistant Enterococcus faecium clones in Denmark, 2015-2023)

Dear Dr. Louise Roer:

Your manuscript has been accepted, and I am forwarding it to the ASM production staff for publication. Your paper will first be checked to make sure all elements meet the technical requirements. ASM staff will contact you if anything needs to be revised before copyediting and production can begin. Otherwise, you will be notified when your proofs are ready to be viewed.

Sincerely,

Vittal Prakash Ponraj Ph.D
Editor
Microbiology Spectrum